# Differential Transcriptional Regulation of Polymorphic p53 Codon 72 in Metabolic Pathways

**DOI:** 10.3390/ijms221910793

**Published:** 2021-10-06

**Authors:** Bu-Yeo Kim, Seo-Young Lee, Sun-Ku Chung

**Affiliations:** 1Division of KM Data, Korea Institute of Oriental Medicine, 1672 Yuseong-daero, Yuseong-gu, Daejeon 34054, Korea; buykim@kiom.re.kr; 2Division of KM Science Research, Korea Institute of Oriental Medicine, 1672 Yuseong-daero, Yuseong-gu, Daejeon 34054, Korea; 09seoyoung03@kiom.re.kr

**Keywords:** polymorphic p53 codon 72, induced pluripotent stem cell, DNA damage stress, growth arrest, apoptosis, metabolic signaling pathway

## Abstract

p53 is a transcription factor that is activated under DNA damage stress and regulates the expression of proapoptotic genes including the expression of growth arrest genes to subsequently determine the fate of cells. To investigate the functional differences of polymorphic p53 codon 72, we constructed isogenic lines encoding each polymorphic p53 codon 72 based on induced pluripotent stem cells, which can endogenously express each polymorphic p53 protein only, encoding either the arginine 72 (R72) variant or proline 72 (P72) variant, respectively. We found that there was no significant functional difference between P72 and R72 cells in growth arrest or apoptosis as a representative function of p53. In the comprehensive analysis, the expression pattern of the common p53 target genes, including cell cycle arrest or apoptosis, was also increased regardless of the polymorphic p53 codon 72 status, whereas the expression pattern involved in metabolism was decreased and more significant in R72 than in P72 cells. This study noted that polymorphic p53 codon 72 differentially regulated the functional categories of metabolism and not the pathways that determine cell fate, such as growth arrest and apoptosis in cells exposed to genotoxic stress.

## 1. Introduction

p53 is a well-characterized transcription factor that is activated under DNA damage stress and thus regulates the expression of proapoptotic genes *Bbc3/*Puma, *Pmaip1/*Noxa, *Bax/*Bax, *and Tnfrsf10b/*Dr5 as well as the expression of growth arrest genes *Cdkn1a/*p21, *Sfn/*14-3-3σ, etc., to eventually determine cell fate [1,2,3]. The inhibition of cell growth and the removal of damaged cells by p53-dependent target gene regulation have important implications for preventing cellular transformation [4,5]. Frequent mutations of the p53 gene located at residues R175, G245, R248, R249, R273, and R282 found in human cancers do not regulate the p53 target gene expression due to DNA contact or structural alteration as transcription factors [6,7]. This functional alteration of p53 due to these mutations leads to serious problems in maintaining genome stability and cell integrity, resulting in cellular transformation [8,9].

In addition to these mutations, as a frequent polymorphism that may have an impact on p53 function, the p53 protein has an amino acid substitution at codon 72, encoding either arginine 72 (R72) variant or proline 72 (P72) variant. It is predictable that the structure of these polymorphic p53 proteins may be altered by non-conservative polymorphic substitutions, which in turn may affect p53 target gene regulation and confer different properties in p53 function [10]. It is common that each of the p53 proteins encoding R72 or P72 exhibits differences in the degree of expression of the p53 target gene, which indicate different growth arrest and apoptosis functions. Specifically, R72 enhances pro-apoptotic gene expression more effectively than P72, leading to effective apoptosis [11,12,13], whereas P72 is more effective at growth arrest than R72 in promoting the expression of cell cycle arrest genes more effectively [13,14].

To investigate the p53 functional difference, which depends on the polymorphic p53 codon 72, previous studies have primarily been based on an isogenic line that artificially induces each variant transiently transfected into p53 null cancer cells [15,16]. Alternatively, we applied a bacterial artificial chromosome-mediated homologous recombination (BAC-mediated HR) system to induce pluripotent stem cells (iPSCs) derived from human normal fibroblasts with heterozygous p53 codon 72 and then replaced two alleles of the p53 gene locus by the BAC. Thus, each clone endogenously expresses the P72- or R72-p53 protein only [17]. We first compared the expression of some of the p53 target genes, such as *Cdkn1a/p21*, *Mdm2*, *Tnfrsf10b*, and *Bbc3/Puma*, between isogenic lines, assuming that the regulation of polymorphic p53 codon 72-dependent target genes has an effect on the difference in growth arrest or apoptosis functions. In addition, we comprehensively analyzed the overall gene expression pattern to determine the effect of P72 or R72 on the expression of p53 target genes on the state of DNA damage.

This study investigated the functional difference of polymorphic p53 codon 72 by demonstrating whether the expression pattern of the p53 target genes is indicated in growth arrest or apoptosis or differentially regulates metabolic signaling as a representative function of p53 on the endogenous expression system of p53 protein, encoding p53 polymorphic amino acid.

## 2. Results

### 2.1. Expression Pattern of Representative p53 Target Genes

We reported an isogenic line for p53 codon 72, including P72 and R72, established by a BAC-mediated HR system based on the human iPSCs identified as retaining p53 heterozygosity for p53 codon 72 [17]. The BAC construct contains the CMV early enhancer/chicken beta actin (CAG) promoter, an extra intron sequence, a neomycin resistance gene, and a stop codon, which are flanked by two Frt sites at a position corresponding to p53 intron1 (Figure 1A). Thus, without removal of the Frt-flanked sequences, the p53 gene expression did not occur in the allele incorporated by the BAC construct. By inserting BAC DNA into one allele of the p53 gene locus, each clone expressed a p53 protein encoding P72 or R72 from another allele in which it was not inserted and showed a difference in protein migration on SDS-PAGE (Figure 1B). From each line, the mRNA or protein expression levels of each polymorphic p53 showed similar patterns, which means that the possibility of the effect of P72 over R72 quantities could be excluded (Figure 1B). To compare the expression levels of target genes regulated by each polymorphic p53 protein, we treated each isogenic line with doxorubicin as a DNA damage-inducing reagent. Each polymorphic p53 showed no significant difference in the regulation of the target genes by p21, a representative growth arrest protein encoded by *the Cdkn1a* gene; mdm2 for cell survival; and Puma, a representative proapoptotic protein encoded by *the BBC3* gene (Figure 1B). In addition, there was no significant difference in transcriptional regulation of p53 target genes, including *Cdkn1a* or *Tnfrsf10b*, regardless of iPSCs or differentiated cells (Figure 1C). Therefore, it is reasonable to assume that the degree of growth arrest caused by p21 or apoptosis induced by PUMA or Dr5 (encoded by *Tnfrsf10b* gene) does not differ noticeably depending on the polymorphic p53 codon 72. Indeed, a similar expression pattern of target genes by polymorphic p53 codon 72 reflects no significant difference in growth arrest or apoptosis functionally (Figure 1D–G). In general, the function of p53 on growth arrest or apoptosis is known to prevent tumor formation. Some studies have reported that one of the polymorphic p53 codon 72 has been predisposed to tumor formation [18,19,20,21,22], but others have argued that there is no significant association between polymorphic p53 codon 72 and cancer risk [23,24,25,26,27,28,29]. According to these controversies, our results showed no significant difference in polymorphic p53 codon 72 dependent target gene expression, which supports the latter that there is no correlation between polymorphic p53 codon 72 and the risk of cancer in terms of preventing tumor formation through p53-dependent target gene regulation.

### 2.2. Similar Global Expression Pattern between the Polymorphic p53 under DNA Damage Stress

In addition to the expression patterns of representative p53 target genes, we comprehensively identified the transcriptional regulation pattern of other target genes between the polymorphic p53 codon 72. We measured the expression profile of genes changed by doxorubicin in each polymorphic cell type, including p53 null cells. As shown in Figure 2A, 600 genes showing variation in expression between samples by doxorubicin were hierarchically clustered. From this gene expression profile, it was confirmed that the number of genes responding to doxorubicin was dominant in both P72 and R72 cells over p53 null cells. In addition, P72 and R72 showed similar gene expression patterns by doxorubicin whereas p53 null cells showed different patterns. This result indicates that doxorubicin induced several genes through p53 dependence. We then isolated temporally up and downregulated genes by doxorubicin in P72 and R72 cells (Figure 2B) and measured their distribution among cell types. As shown in Figure 2C, unlike p53 null cells, a large number of genes (154 temporally upregulated genes and 9 temporally downregulated genes) in P72 and R72 cells responded in common at 4 h after doxorubicin treatment. We then measured the biological functions associated with genes temporally regulated by doxorubicin. As shown in Figure 2D, genes in p53 null cells were associated with nucleosome assembly whereas genes in P72 or R72 cells were associated with cell cycle arrest, drug response, and DNA damage response. This result is consistent with the fact that there were many common genes between the genes isolated from the two cells (Figure 2B) and proves the role of doxorubicin as a DNA damaging agent.

### 2.3. Patterns of Cellular Signaling Pathways between the Polymorphic p53 Codon 72

This time, the biological function change by doxorubicin was measured by directly measuring the pathway activity. Pathway activities were calculated by measuring the cumulative effect of the expression of genes included in each pathway. As shown in Figure 3A, many pathways were upregulated by doxorubicin in p53 null cells, but there were a number of pathways that were downregulated in P72 and R72 cells. The overall pattern of pathway activity showed a close correlation (r = 0.77, *p* < 0.001) between P72 and R72 cells but a low correlation between P72 (or R72) and p53 null cells (Figure 3B). Categorical classification of pathways showed that, in P72 or R72 cells, pathways included in the metabolic and genetic information processing categories were downregulated whereas pathways in the signaling category were upregulated by doxorubicin. For example, the activities of the citrate cycle and apoptosis pathway were significantly downregulated and upregulated, respectively, in both the P72 and R72 cells when compared with p53 null cells. Although the overall pathway activity distribution was similar between P72 and R72 cells, more detailed pathway category analysis confirmed that some specific functions were different between P72 and R72 cells. In particular, it was determined that the activity of pathways involved in metabolic function was lower in R72 (Figure 3C).

### 2.4. Neighbor-Based Isolation of Doxorubicin Responsive Genes

It is possible that the functional difference induced by a single nucleotide change between P72 and R72 types of p53 does not significantly contribute to the main functional changes induced by doxorubicin. Specifically, the polymorphic p53 codon 72 may not respond differently to DNA damage reagents depending on P72 or R72, or it may be difficult manifest itself because it is obscured by major functional changes. However, considering the limitations of experimental conditions, there are not many options other than using an efficient p53 inducer such as doxorubicin. In addition, increasing the number of replicates for the isolation of detailed functional changes is difficult because of cost issues. Therefore, we applied a novel analytical approach that used interaction networks of genes to isolate genes differentially regulated between two experimental conditions. Specifically, a module consisting of the first neighbor interacting with each gene of interest was isolated from the integrative interaction network. Then, the expression levels of the genes included in the module were used to determine the expressional differences, expressional deviations, and correlation coefficients between the two experiments, and the statistical significance was determined by comparing it with the results from the randomly selected modules from two experiments. Finally, modules showing empirical *p* < 0.01 were isolated, and the core genes of each module were called core differentially expressed genes (core DEGs).

As shown in left panel of Figure 4A, 325 and 320 core DEGs were identified between p53 null cells and P72 or R72 cells at 4 h after doxorubicin treatment, respectively. The high number of common core DEGs (144 genes) among them indicates similarities between P72 and R72 cells compared with p53 null cells. On the other hands, 129 and 339 core DEGs were isolated between the P72 and R72 cells at 2 h and 4 h after doxorubicin treatment, respectively (right panel of Figure 4A). The increase in core DEGs (from 129 to 339 genes) according to doxorubicin treatment time and a relatively small number of common core DEGs (24 genes) among them showed that the difference between P72 and R72 increased with time after doxorubicin treatment. The selected list of core DEGs is shown in Table 1.

Measurement of the functional associations for these core DEGs between p53 null cells and P72 or R72 cells at 4 h after doxorubicin treatment showed that p53-dependent functions such as cell cycle, apoptosis, and DNA damage response were significantly enriched by doxorubicin treatment (Figure 4B). These results are consistent with the generally known function of p53 on doxorubicin, thus demonstrating that the network-based gene selection method used in this study could be very useful for measuring biological changes. In contrast, core DEGs between p72 and R72 cells, which represent genes differentially expressed between P72 and R72 cells in response to doxorubicin (4 h after treatment) were associated with metabolism, particularly lipid-related metabolism (false discover rate (FDR) < 0.01) in pathway enrichment analysis as shown in upper panel of Figure 4C. Lipid metabolism-related GO-terms were also enriched (FDR < 0.01) in the GO-term network analysis, as shown in lower panel of Figure 4C.

We then measured the distribution in the whole interaction network of core DEGs isolated between P72 and R72 cells in response to doxorubicin (4 h after treatment). As shown in the left panel of Figure 4D, the core DEGs were distributed across the entire area of the interaction network. Following the measurement of interactions between these core DEGs, it was observed that a number of genes (261 out of 339) formed a cluster in which they interact directly, as shown in the right panel of Figure 4D. Interestingly, this cluster contained p53 as a core DEG. This result indicates that the genes involved in the difference in response to doxorubicin according to the p53 status were closely related to each other in the network centered on p53.

### 2.5. Distances in Network between Metabolism, p53, and DNA Damage

For quantitative analysis of the functional relatedness of core DEGs, we measured the distances in the network of genes with pathways. Previously, we reported that the similarity in distances between genes in the network is related to the similarity of functions [30]. For example, genes involved in metabolic pathways are distributed in the interaction network from genes involved in signaling pathways. Therefore, using the correlation between the distance of genes and its function, pathways with similar distance distributions on the network with core DEGs were selected. Figure 5A shows that the metabolic pathways were distinct from the signaling pathways in the distance distribution on the network, consistent with the results of our previous report. The core DEG that differentially responds to doxorubicin between P72 and R72 cells showed a distance distribution pattern similar to that of metabolic pathways rather than signaling pathways. For a more detailed analysis, the metabolic pathway cluster was only selected, and the distance of distribution was measured again. The core DEGs showed a very similar distribution to pathways such as steroid biosynthesis, cholesterol metabolism, terpenoid backbone biosynthesis, ferroptosis, and glucagon signaling pathway, as shown in Figure 5B. The activity levels for these selected pathways were measured as indicated in the left panel of Figure 5C and the relative ratio of pathway activity changes between p72 and R72 was measured in the right panel of Figure 5C. In all six pathways, at least 50% or more changes were observed in P72 cells compared with R72 cells.

### 2.6. Analysis of External Public Datasets

We then measured the biological differences between the P72 and R72 forms of p53 using three publicly available external datasets and compared the results obtained from our study. The first (GSE61124) observed gene expression changes induced by gamma ray radiation on the P72 and R72 forms of p53 in human fibroblasts [31], and the second (GSE109373) measured the effects of P72 and R72 forms of p53 in mutated P53 R175H and R273H) human non-small cell lung carcinoma cell line [32]. The last (GSE26851) was genome expression data obtained by irradiating ionizing radiation on P72 form or R72 form thymocytes isolated from humanized p53 knock-in (Hupki) mice [33].

We first measured the pathway activity profile of the public samples in comparison with this study. As shown in Figure 6A, most of the metabolic pathways were downregulated by external stimuli in P72 or R72 samples. The pathway activity-based correlation plot supports this overall similarity between samples (Figure 6B). However, since it was difficult to distinguish the difference between the P72 and R72 forms of p53 from external data in terms of pathway activity, the abovementioned method was used to select the core DEGs that show differential expression between P72 and R72 forms of p53. Figure 6C shows the distribution of these core DEGs between the external samples and our sample. There were some core DEGs in common between our experimental sample and external data but not many. Therefore, this time, we measured whether the functions associated with core DEGs are similar between samples. As shown in Figure 6D, metabolism-related pathways, particularly lipid- or amino acid-related metabolic pathways, were commonly associated with core DEGs in the external and our datasets. In the GSE26851 data, a small number of functions were associated with core DEGs, including arachidonic acid metabolism, NF-kappa B signaling pathway, and cytokine receptor interaction pathway. Interestingly, these functions were also mentioned by the authors [33]. These results show that functional analysis using core DEGs can be useful in identifying biological differences despite differences in individual genes.

## 3. Discussion

The polymorphic codon 72 residue located in human *TP53* exon 4 encodes a frequent arginine or proline amino acid. Since the relationship of each amino acid has a non-conservative polymorphic substitution that can confer a structural difference, it can be assumed that they will also induce differences in growth arrest and apoptosis as the typical functions of p53. If the functional difference of growth arrest or apoptosis depends on the different polymorphic p53 codon 72, it can be assumed that P72 or R72 suppresses tumor formation by regulating major cell cycle arrest and minor apoptosis or a minor cell cycle arrest or major apoptosis. Some studies have observed no association between the polymorphic p53 codon 72 and cancer prevention, which support this assumption [23,24,25,26,27,28,29]. Conclusively, each polymorphic p53 codon 72 mice retaining P72 or R72, respectively, was not able to determine which type was more prone to cancer but also showed no difference in survival rate [33]. Exploring the correlation with the expression of p53 target genes depending on polymorphic types in the mouse-derived embryonic fibroblast, the expression of *Cdkn1a*, a gene related to growth arrest, was relatively increased by P72 [33]. This result suggests cell cycle arrest that is more strongly regulated by P72, consistent with previous results [13,14]. However, incompatible with previous reports [11,12,13], there was no significant difference in the expression of proapoptotic genes including Puma, Noxa, and Bax by R72, comparing with P72 [33].

On the other hand, some studies have reported that one of the polymorphic p53 codon 72 has been predisposed to tumor formation [18,19,20,21,22]. R72 was enriched in human papilloma virus-positive cervical cancers [18,19,20]. It was also significantly associated with an increased risk of developing melanoma [21]. In addition to R72, P72 is known to induce tumor promotion by increasing cell survival and epithelial-mesenchymal transition and consequently contributing to the aggressiveness of colorectal cancer, inducing activation of the p38 and RAF/MEK/ERK pathways [22]. The phenotypes have been reflected in the regulation of growth arrest genes or pro-apoptotic genes as p53 target genes. As aforementioned, many previous reports have suggested that P72 strongly regulates growth arrest, whereas R72 induces more apoptosis than cell cycle arrest. However, in this study, we did not find a significant difference in the representative functions of p53 between P72 and R72. Unlike previous studies, our results showed a similar expression pattern of p53 target genes related to cell cycle arrest or apoptosis as well as similar phenotypes, including growth arrest or apoptosis. These results can be attributed not only to tissue-specific differences but also to different experimental models. In particular, while most of the previous studies primarily compared the differences by artificially expressing each polymorphic p53 codon 72 vector in the p53 null cancer cell lines, our study comparatively analyzed isogenic lines capable of endogenously expressing either P72 or R72, based on induced pluripotent stem cells generated from normal human fibroblasts. Although only another functional copy is retained by removing one copy of the polymorphic p53 codon 72, the retention of another copy can apparently reflect the functional differences between isogenic lines. Additionally, our isogenic line retains heterozygosity of p53; thus, it may provide valuable research for Li-Fraumeni syndrome, including loss of heterozygosity at the p53 gene locus, as an inherited p53 mutation.

The polymorphic p53 codon 72 has a characteristic as a non-conservative single nucleotide change, but the functional similarity between P72 and R72, such as cell cycle arrest and apoptosis, is so close that it was somewhat difficult to clearly identify the functional difference according to the polymorphism. Therefore, we applied the gene interaction network, the so-called network-based gene selection method, to isolate genes differentially regulated by the polymorphic p53 codon 72. The useful value of this analytical method was confirmed by the apparent functional difference between p53 null and polymorphic p53 codon 72, such as cell cycle arrest, apoptosis, or DNA damage response. Through the network-based gene selection method, we could identify that core DEGs representing genes differentially expressed between P72 and R72 cells in response to doxorubicin were associated with metabolism, particularly lipid-related metabolism.

R72 is known to be associated with metabolic diseases compared with P72. In a recent study of humanized mice carrying the polymorphic p53 codon 72, Kung et al. reported that R72 mice are predisposed to obesity and metabolic dysfunction. The R72 mouse had the most noticeable difference at the beginning of the high-fat diet, which was obesity, compared with the P72 mouse [34]. The severe obesity phenotype developed into glucose intolerance, insulin resistance, islet hypertrophy, and fatty liver disease. According to the gene expression analysis results between the two mice, the difference in phenotypes for metabolic diseases was due to the differences in the expression of *Tnf* and *Npc1I1* genes involved in inflammation and cholesterol metabolism, respectively. The reason we could not confirm the difference in the expression of the two genes in the P72 and R72 cell lines can be seen as a result of the difference in experimental conditions. That is, unlike our genotoxic stress, the previous study produced data from HFD-fed mouse. The results of experiments in mice carrying humanized polymorphic p53 codon 72 support the existing cohort study result that the actual polymorphic p53 codon 72 is linked with susceptibility to diabetes [35]. In this context, R72 was associated with increased insulin resistance [36]. In addition, waist circumference and obesity have been reported to be closely related to R72 as risk factors for type 2 diabetes.

Similarly, previous studies on the association of R72 with metabolic diseases were identified under non-genotoxic stress, but this may lead to changes in the metabolic functional category by R72 under genotoxic stress. This study confirmed that the expression pattern of metabolism-related genes was altered under genotoxic stress, a stimulant severe enough to determine cell fate. As shown in Figure 2B, the expression of metabolism-related genes was significantly reduced in both P72 and R72 cells under DNA damage conditions, and in particular, the expression of these genes showed a tendency to further decrease in R72 cells. Although further experimental confirmation is required, this suggests that metabolic dysfunction may be relatively vulnerable in R72 cells through the reduction of metabolism-related genes, which is more marked in R72 than in P72.

## 4. Materials and Methods

### 4.1. Cell Culture

Human iPSCs, including P72, R72, and p53 null cells, were established and maintained as previously reported [17]. Human fibroblasts were reprogrammed using episomal vectors to express Oct4, Sox2, Klf4, Lin28, and L-myc using the Neon Transfection System (Invitrogen, Carlsbad, CA, USA). Human iPSCs were cultured in mTeSR1 medium (STEMCELL Technologies, Vancouver, BC, Canada) on a Matrigel-coated cell culture dish or feeder layer. ReLeSR^TM^ (STEMCELL Technologies) was used to passage hiPSCs.

### 4.2. Protein Extraction and Western Blotting

Total cellular protein was extracted from human iPSCs using NP40 cell lysis buffer (Invitrogen) supplemented with PIC and PMSF. 20 ug of the cell lysates were separated on an SDS-PAGE gel and transferred to nitrocellulose membranes. Blotted membranes were blocked with 5% skim milk in TBST for 1 h, followed by incubation with specific primary antibodies. For the detection of proteins, membranes were probed with primary antibodies against p53 (Cell Signaling Technology, #48818, Danvers, MA, USA), p21 (Cell Signaling Technology, #2947), mdm2 (Cell Signaling Technology, #86934), PUMA (Santa Cruz Biotechnology, #sc-374223, Dallas, TX, USA), Phospho-H2AX (Cell Signaling Technology, #9718), and ꞵ-actin (Santa Cruz Biotechnology, #sc-47778). HRP-conjugated anti-rabbit (ENZO, #ADI-SAB-300-J, New York, NY, USA), and anti-mouse (ENZO, #ADI-SAB-100-J) were used as secondary antibodies. Immunoreactive bands were visualized using ImageQuant LAS 4000 mini (GE Healthcare, Chicago, IL, USA).

### 4.3. RNA Extraction and Quantitative Real-Time PCR

Total RNA was extracted from the cell pellets using the RNeasy Mini Kit (QIAGEN, Hilden, Germany) according to the manufacturer’s protocol. Reverse transcription was performed using the SuperScript III First-Strand Synthesis System (Invitrogen). Synthesized cDNA was diluted and then quantified with gene-specific primers using Power SYBR Green PCR Master Mix (Applied Biosystems, Foster City, CA, USA). Real-time PCR was performed using a QuantStudio 6 Flex Real-time PCR System (Applied Biosystems). *GAPDH* was used as an endogenous control.

### 4.4. Cell Growth Arrest Assay

To quantify the DNA synthesis rate between the P72, R72, and p53 null cells, endodermal lineage differentiation was induced from each type of human iPSCs using a STEMdiff™ Definitive Endoderm Kit (STEMCELL Technologies). Before the incorporation of 5-ethynyl-2’-deoxyuridine (EdU) into the synthesized DNA, 0.2 μM doxorubicin was treated on the differentiated cells. Twelve hours later, cells were incubated overnight with 10 μM EdU-containing cell culture medium in accordance with the manufacturer’s protocol using a Click-iT Plus EdU imaging kit (Invitrogen). Images were obtained using an Olympus IX71 inverted fluorescence microscope (Olympus, Tokyo, Japan).

### 4.5. Annexin V Analysis

To examine the apoptosis of human iPSCs induced by DNA damage, cells were seeded on a 6-well cell culture plate and pre-treated with 0.2 μM doxorubicin for 6 h. Cells were harvested and washed with ice-cold PBS and stained using the BD Pharmingen™ Annexin V-FITC Apoptosis Detection Kit (BD, Franklin Lakes, NJ, USA) according to the manufacturer’s protocol. Annexin V-FITC and propidium iodide (PI)-stained cells were detected using an LSRFortessa X-20 cell analyzer (BD Bioscience), and the data were analyzed using FlowJo software (FlowJo, Ashland, OR, USA).

### 4.6. Statistical Analysis

The data are presented as mean ± SD from at least three independent experiments. Student’s *t*-test was used to determine statistical differences. Statistical significance was set at *p* < 0.05.

### 4.7. RNA Isolation, Library Preparation, and Sequencing for QuantSeq Analysis

Total RNAs were purified from iPSCs using Trizol according to manufacturer instructions (Thermo Fisher Scientific, Rockford, IL, USA). After quality assessment of the RNA by 2100 Bioanalyzer Instrument (Agilent, Santa Clara, CA, USA), only samples with an RNA integrity number >7.0 were included in the QuantSeq analysis. From 500 ng total RNA, double-stranded cDNA library was prepared using QuantSeq 3′ mRNA-Seq Library Prep Kit (Lexogen, Vienna, Austria), amplified by polymerase chain reaction, and then subjected to sequencing using Illumina NextSeq 500 system (Illumina, San Diego, CA, USA).

### 4.8. QuantSeq Data Analysis

QuantSeq reads were aligned to a reference genome (genome assembly version of hg38) using Bowtie2 [37] to estimate the amount of transcripts. Read count data were then normalized to obtain fragments per kilobase of million reads mapped (FPKM) using the edgeR R package [38].

### 4.9. Clustering of Genes and Isolation of Temporally Regulated Genes

After selecting genes in which the change in expression ratio compared with the control was more than 0.1 standard deviation from among the samples, the genes were hierarchically clustered according to the expression ratio using the Gene Cluster 3.0 program and visualized using the Java TreeView program [39]. Genes regulated by time patterns were isolated using the Short Time-series Expression Miner (STEM) program with the statistical significance of the patterns set to FDR < 0.01 [40].

### 4.10. Construction of Interaction Network

Information on physical interaction between proteins were obtained from BioGRID (BIOGRID-ORGANISM-Homo sapiens version 3.5, https://thebiogrid.org/, accessed on 1 October 2021) [41]. Information on proteins interacting with each other while sharing metabolites was obtained from the entry of compounds with biological roles in the KEGG COMPOUND Database (https://www.genome.jp/kegg/compound, accessed on 1 October 2021). By integrating this database information, an integrated interaction network was constructed.

### 4.11. Isolation of Core DEGs

For each experiment, a module composed of individual gene and its first neighbors were isolated and its average and standard deviation in gene expression were measured from an integrated interaction network. Between experiments, the correlation coefficient of gene expression was also measured. Statistical significances for the measured statics were determined empirically by iterating this process 1000 times with randomly selected genes in integrated network.

### 4.12. Functional Enrichemtbn of GO-Terms

GO-terms network were constructed by implementing the ClueGO [42] Cytoscape plugin application (downloaded from http://www.ici.upmc.fr/cluego, accessed on 1 October 2021). Input genes selected from experiments were queried into ClueGO by using the following parameter settings: Benjamin–Hochberg FDR < 0.01, kappa score > 0.4 and GO terms fusion check.

### 4.13. Pathway Activity

The pathway activity was examined by calculating the cumulative effect of the expression of genes included in each pathway [30,43]. In brief, the log-transformed expressional ratios of genes relative to the control in each pathway were added linearly with a weight of −1 for genes that acted as repressors, which were defined as proteins that inhibited the process of signal transduction of the pathway. The measured value of pathway activity was normalized by dividing it by the size of the pathway. Statistical significance was empirically calculated by comparing the measured values with those obtained from a random permutation of 1000. Information on individual pathways and their functional categories was obtained from the Kyoto Encyclopedia of Genes and Genomes database (KEGG PATHWAY, http://www.genome.jp/kegg, accessed on 1 October 2021).

### 4.14. Measurement of Distance between Genes in the Network

Associations between sets of genes were measured using distances on the network. The shortest path was defined as the distance between two genes in the network. More specifically, the igraph R package (version 1.2.6) was used to map each gene in one gene set to an interaction network to determine the shortest path to each gene in another gene set. After repeating this process for all gene pairs, we constructed a matrix of shortest paths from both gene sets. Then, the mean of the shortest path matrix was taken as the distance between the two sets of genes. For the statistical evaluation of the measured distance values, this process was repeated 1000 times with a randomly selected set of genes with the same number of genes to adjust for the effect of the size of the gene set on the distance distribution.

## Figures and Tables

**Figure 1 ijms-22-10793-f001:**
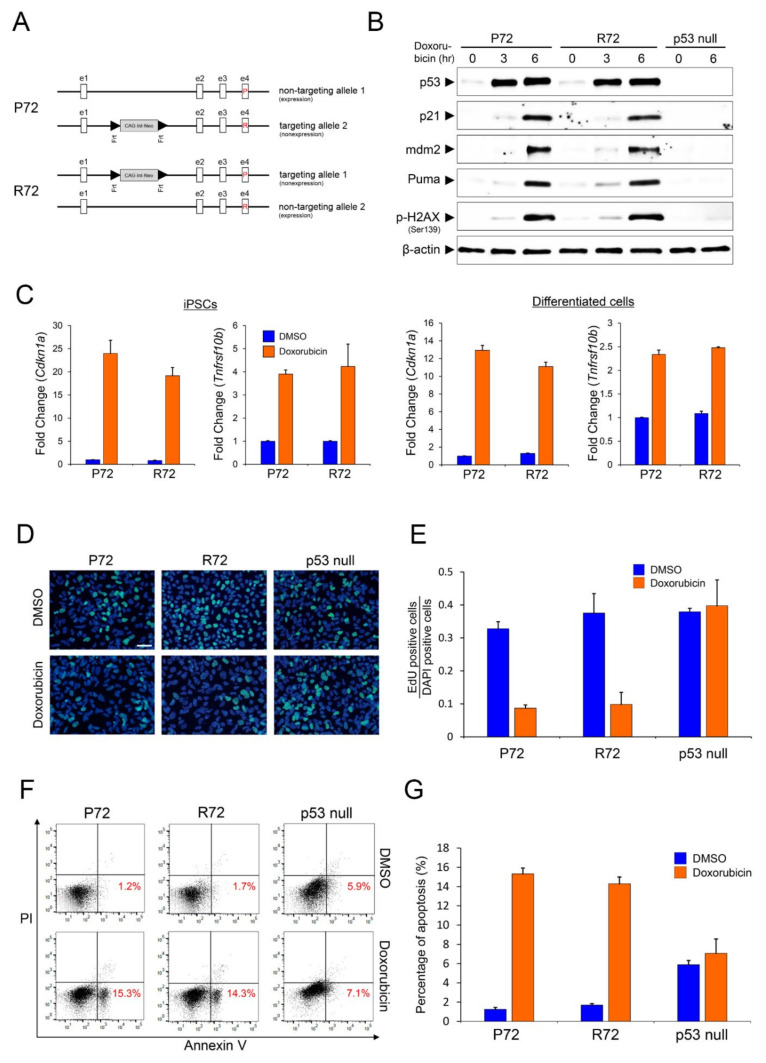
Similar functional patterns shown in P72 and R72 cells under DNA damage stress. (**A**) Targeting scheme for polymorphic p53 codon 72. The parent cells have an inherited heterogenous polymorphism at codon 72 at TP53 exon 4. Each allele was used in the construction of bacterial artificial chromosome (BAC)-based targeting vectors (**B**) Western blot analysis of P72, R72, and p53-dependent target genes in P72, R72 and p53 null cells. The cells were either untreated or doxorubicin (0.2 µM) treated for 3 h and 6 h. The blot was probed with total p53, p21, mdm2, Puma, and p-H2AX antibodies. β-actin was used as a loading control. (**C**) mRNA levels of *Cdkn1a* and *Tnfrsf10b* in DMSO-treated or doxorubicin (0.2 µM)-treated iPSCs and differentiated cells of each P72 and R72 cell were measured by Real-time PCR. The results are expressed as mRNA fold change relative to that of DMSO treated cells. The data represent mean ± standard deviation from three independent experiments. (**D**,**E**) Cell growth arrest assay in the endodermal cell lineage derived from P72, R72, and p53 null-human iPSCs. Cells were DMSO or 0.2 µM doxorubicin treated for 12 h and then pulsed with 10 µM EdU for 16 h. EdU-positive cells (green) and DAPI-positive cells (blue) were evaluated by immunofluorescence analysis. Scale bar = 100 *µ*m. The data in (**E**) represent mean ± standard deviation from three independent experiments. (**F**,**G**) Apoptosis of P72, R72, and p53 null cells treated with DMSO or 0.2 µM doxorubicin for 6 h were assayed by Annexin V-FITC and PI staining. Stained cells were measured by flow cytometry analysis. (**F**) Mean values from three independent experiments are shown with the standard derivation. (**G**) The data represent mean ± standard deviation from three independent experiments.

**Figure 2 ijms-22-10793-f002:**
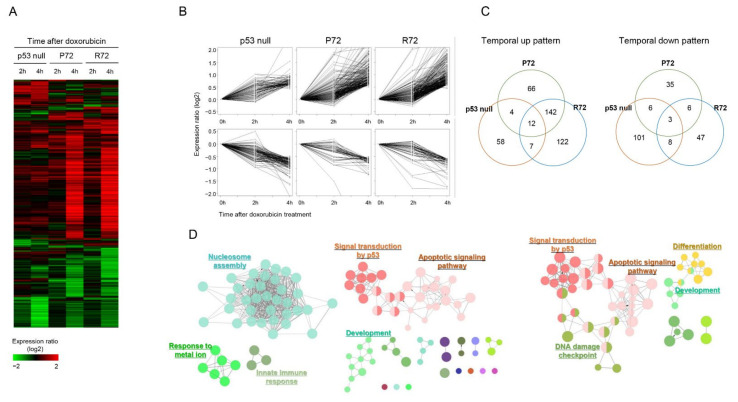
Effect of p53 status on doxorubicin-induced gene expression: (**A**) 600 genes showing an expression change of at least 0.1 standard deviation in samples were clustered according to the expression level. Red and green color represents high and low expression level, respectively, as shown in the scale bar. (**B**) Genes showing temporal pattern of expression were isolated in P72, R72, and p53 null cells. (**C**) Distribution of temporally regulated genes were compared among P72, R72, and p53 null cells. (**D**) Association of temporally regulated genes from P72, R72, and p53 null cells with the GO term was measured using the ClueGO program to make a GO-term network. Each node represents a GO term, and the node size is proportional to the number of genes associated with the GO term. Closely related GO terms are marked with the same color. Representative GO terms are highlighted.

**Figure 3 ijms-22-10793-f003:**
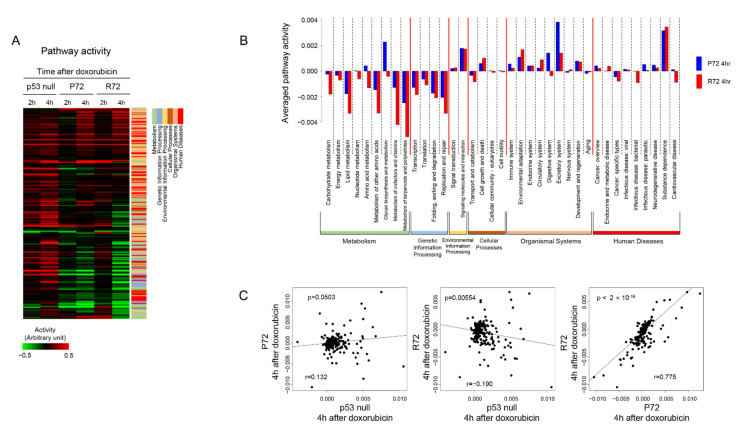
Effect of p53 status on doxorubicin-induced pathway activity. (**A**) Pathways were clustered according to the activity level in samples. Red and green indicate high and low activity levels, respectively, as indicated by the scale bar. Pathways were classified into six functional categories as indicated by color below. (**B**) Mean activity of pathways was measured according to functional category in P72 and R72 cells. (**C**) P72, R72, and p53 null cells were compared according to pathway activity distribution.

**Figure 4 ijms-22-10793-f004:**
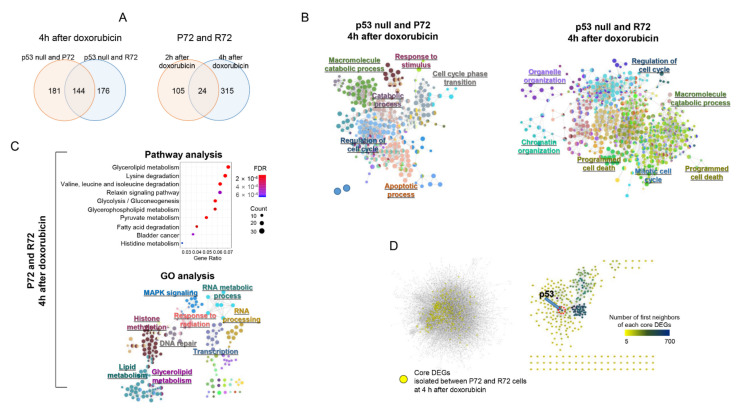
Functional difference among P72, R72, and p53 null cells. (**A**) After selecting the core DEGs from each of P72, R72, and p53 null cells, their distributions were compared. (**B**) The association of core DEGs between P72 (or R72) and p53 null cells with GO-term was measured using the ClueGO program to make a GO-term network. Each node represents a GO term, and the node size is proportional to the number of genes associated with the GO-term. Closely related GO-terms are marked with the same color. Representative GO-terms are highlighted. (**C**) Functional associations of core DEGs between P72 and R72 cells were measured in pathway (upper panel) and GO-term enrichment assays (lower panel). In the pathway analysis (upper panel), the distribution of false discovery rate (FDR) values of each pathway and the number of genes included in the pathway were shown according to the ratio of genes included in the pathway. In the GO-term analysis (lower panel), the ClueGO program was used to construct an enriched GO-term network. (**D**) The location of each core DEG between the P72 and R72 cells was marked in the interaction network. Directly connected core DEGs were indicated in the right panel, and the color of the nodes indicates the number of nodes connected to each node, as shown in scale bar. The positon of p53 genes was indicated.

**Figure 5 ijms-22-10793-f005:**
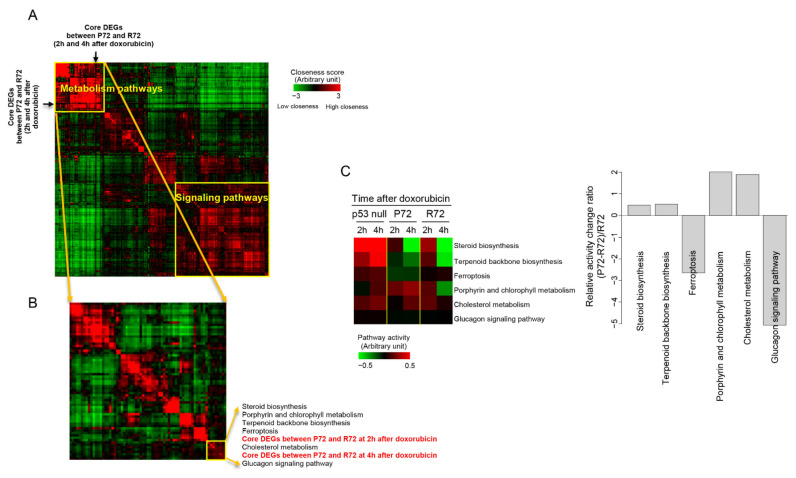
Measurement of the distance in the network. (**A**) Pathways were clustered according to distance distribution patterns of genes included in the pathways in the network. The location of core DEGs that were differentially expressed between P72 and R72 cells is indicated. (**B**) One cluster composed mainly of metabolic pathways and core DEGs between P72 and R72 cells was isolated and re-clustered. A cluster containing the core DEGs between P72 and R72 cells was finally isolated, and the pathways comprising this cluster are shown. (**C**) In the left panel, the activity levels for these finally isolated pathways are color-coded. Red and green indicate high and low activity levels, respectively, as indicated by the scale bar. In the right panel, the relative ratio of pathway activity changes between p72 and R72 was measured for these selected pathways. The relative ratio of pathway activity was determined as (P72-R72)/R72.

**Figure 6 ijms-22-10793-f006:**
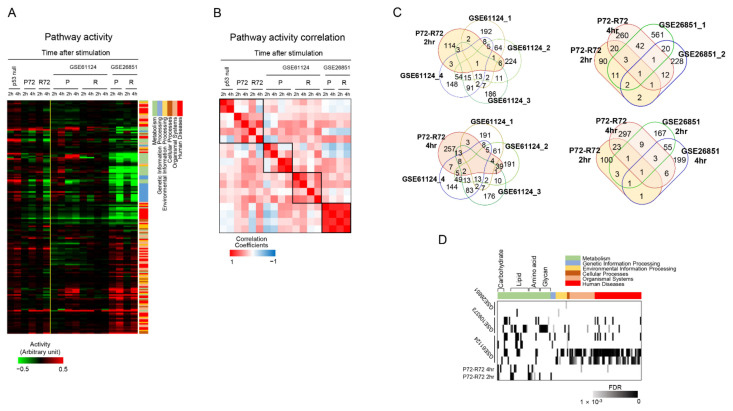
Analysis of external public datasets. (**A**) Pathway activity from external datasets and our samples were measured and clustered. Red and green indicate high and low activity levels, respectively, as indicated by the scale bar. Six functional categories including each pathway are indicated by color next to the figure. In GSE61124, P72 or R72 was present for each of the two cell types, respectively. (**B**) Correlation among samples were measured according to activity level of pathway. Red and blue indicate high and low activity levels, respectively, as indicated by the scale bar. (**C**) The distribution of core DEGs selected from the external data sets was compared with those selected from our P72 and R72 cells. In GSE61124, four types of core DGEs were isolated from P72 and R72 at 4 h after stimulation for each cell type, and in GSE26851, two types of core DGEs were isolated for each cell type. The core DEGs in our experiments are highlighted in color. (**D**) Pathways associated with core DEGs selected from external datasets were compared between samples. Intensity in black indicates statistical significance as shown in the scale bar below.

**Table 1 ijms-22-10793-t001:** Selected list of core DEGs. The top 50 genes are displayed in order of number of the first neighbor of a given gene in the interaction network. Common genes between key DEGs were shown; “null-P” represents core DEGs between p53 null and P72 cells. “null-R” represents core DEGs between p53 null and R72 cells. “P-R” represents core DEGs between P72 and R72 cells.

p53 Null-P72 4 h after Doxorubicin				p53 Null-R72 4 h after Doxorubicin				P72-R72 4 h after Doxorubicin			
Symbol	# of First Neighbors	Empirical *p*-Value	Common Gene	Symbol	# of First Neighbors	Empirical *p*-Value	Common Gene	Symbol	# of First Neighbors	Empirical *p*-Value	Common Gene
TP53	684	0.001	null-R P-R	MYC	908	<0.001		TP53	684	<0.001	null-P null-R
UBC	383	0.003	null-R	TP53	684	<0.001	null-P P-R	POLR2A	291	<0.001	
MDM2	347	<0.001	null-R	UBC	383	<0.001	null-P	U2AF2	249	0.007	null-R
AR	327	<0.001	null-R	MDM2	347	<0.001	null-P	H2AX	212	<0.001	null-R
SP1	272	<0.001	null-R	AR	327	<0.001	null-P	SIRT1	204	0.001	null-R
CREBBP	268	<0.001	null-R	HDAC1	299	0.003		ACOT8	193	<0.001	
RELA	243	<0.001	null-R	SP1	272	<0.001	null-P	HSPA4	192	<0.001	
JUN	232	0.004	null-R	CREBBP	268	0.003	null-P	DLD	190	<0.001	
HIF1A	228	0.001	null-R	U2AF2	249	0.001	P-R	ACAA2	187	<0.001	
NFKB1	201	0.001	null-R	RELA	243	0.001	null-P	KMT2A	171	<0.001	
SMAD3	179	0.001	null-R	JUN	232	0.001	null-P	STUB1	160	<0.001	null-R
SQSTM1	178	0.001	null-R	HIF1A	228	0.001	null-P	HSPA8	152	<0.001	
BTRC	172	0.001	null-R	UBE2I	228	0.001		PRMT5	144	<0.001	
NR3C1	165	0.001	null-R	H2AX	212	0.001	P-R	HRAS	137	<0.001	
RHOA	159	0.001	null-R	SIRT1	204	0.001	P-R	CREB1	135	<0.001	
STAT3	158	0.002	null-R	H3C1	203	0.001		LPCAT1	129	<0.001	
UBE2D1	155	0.004	null-R	NFKB1	201	0.002	null-P	STAT1	119	<0.001	
CUL3	152	0.002	null-R	PCNA	182	0.002		KMT2D	119	<0.001	
ITCH	148	0.002	null-R	SMAD3	179	0.002	null-P	POLR1A	119	<0.001	
UBE2D3	146	0.003	null-R	SQSTM1	178	0.002	null-P	HADHB	118	<0.001	
CBX5	135	0.003	null-R	BTRC	172	0.002	null-P	ACAA1	115	<0.001	
CBL	132	0.01	null-R	RB1	170	0.004		SF3A2	114	<0.001	null-R
RBX1	130	0.003	null-R	VCP	169	0.003		NCOA1	114	<0.001	
ABL1	128	0.003	null-R	E2F2	167	0.003		SETD1A	114	<0.001	
RARA	125	0.003	null-R	NR3C1	165	0.042	null-P	TERF1	112	<0.001	
TRAF2	122	0.007	null-R	STUB1	160	0.042	P-R	KDM2A	111	<0.001	
CDK9	114	0.003	null-R	RHOA	159	0.004	null-P	PCK2	108	<0.001	
ETS1	107	0.003		STAT3	158	0.004	null-P	KMT2B	106	<0.001	
FZR1	105	0.001	null-R	UBE2D1	155	0.004	null-P	KMT5C	106	<0.001	
CHEK2	104	0.004	null-R	CUL3	152	0.004	null-P	PLOD1	106	<0.001	
XIAP	93	0.004		PML	149	0.006		LPCAT4	105	<0.001	
AURKA	92	0.004	null-R	ITCH	148	0.004	null-P	SMYD3	105	<0.001	
UBE2N	88	0.004	null-R	UBE2D3	146	0.005	null-P	KRAS	104	<0.001	null-R
TSG101	88	0.004	null-R	UBE2D2	139	0.005		PLOD2	104	<0.001	
SPRTN	84	0.002	null-R	CBX5	135	0.005	null-P	CDC20	103	<0.001	null-R
FBXW11	77	0.002	null-R	CBL	132	0.005	null-P	RIOX1	103	0.002	
PTK2	76	0.005	null-R	RBX1	130	0.005	null-P	ALDH3A2	103	0.001	
BCL2L1	75	0.005		ABL1	128	0.002	null-P	ADSS2	101	0.003	
NBN	74	0.005	null-R	ATM	127	0.002		ALDH2	101	0.001	
UBE2E1	72	0.005	null-R	RARA	125	0.002	null-P	PARK7	100	0.001	
CENPA	68	0.005	null-R	TRAF2	122	0.006	null-P	ALDH7A1	100	0.001	
SUMO2	68	0.005	null-R	SF3A2	114	0.006	P-R	PIN1	99	0.009	null-R
EEF1A1	68	0.004	null-R	SMAD4	114	0.002		KMT5B	99	0.001	
SCP2	68	0.001	P-R	TBP	112	0.007		ALDH9A1	99	0.003	
FANCD2	67	0.005	null-R	ETS1	107	0.007	null-P	ALDH1B1	98	0.001	
CCNA2	64	0.008	null-R	FZR1	105	0.003	null-P	SETD5	97	0.004	
NOTCH1	60	0.006	null-R	CHEK2	104	0.002	null-P	WDR5	91	0.001	
KEAP1	57	0.006		KRAS	104	0.008	P-R	HSPA1A	89	0.001	
CERS4	56	0.006	null-R	CDC20	103	0.009	P-R	PDHA1	87	0.004	
CERS5	56	0.002	null-R	RXRA	99	0.003		APP	86	0.002	

## Data Availability

All data associated with this study are available upon reasonable request. The RNA-seq datasets generated and analyzed in this study will be available in the Gene Expression Omnibus (GEO) repository under accession number GSE183668.

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
