# Peer review of "Differential Transcriptional Regulation of Polymorphic p53 Codon 72 in Metabolic Pathways"

_ijms, 2021, doi:10.3390/ijms221910793_

Round 1
Reviewer 1 Report
Dear authors,
This manuscript investigates the functional differences of polymorphic p53 codon 72, in isogenic lines encoding each polymorphic p53 codon 72 based on induced pluripotent stem cells.
Overall, the manuscript is informative to readers, authors reported new information on global expression pattern of representative p53 target genes, polymorphic p53 under DNA damage stress, cellular signaling pathways between the polymorphic p53 codon 72, doxorubicin responsive genes, network between metabolism, p53, and DNA damage and support of their results by published studies. This study well aligns with in vivo study published by Kung et al 2017 in human p53 knock-in (Hupki) mice that suggests R72 polymorphism confers increased cell survival in response to nutrient deprivation, it holds true for the northern hemisphere where people are deprived of nutrients because of scarcity, it shows biased selection of R72 of nature. However, conclusion about differentially regulation of metabolic function by polymorphic p53 codon 72 is not new. There are several cancer research related articles published show different regulation pathways of DNA repair and apoptosis. Hu et al 2005 reported Arg/Pro polymorphism at codon 72 of the p53 gene alters the ability of the p53 protein to induce apoptosis, influences the behavior of mutant p53, decreases DNA repair capacity, and may be linked with an increased risk of lung cancer. I suggest authors should include in vivo data of Hupki mice to support their results about codon 72 polymorphism of p53 does not regulate DNA damage, growth arrest and apoptosis pathway if they are stand by their results in iPSc.
English needs to improve in the entire manuscript.
Thank you
Author Response
This manuscript investigates the functional differences of polymorphic p53 codon 72, in isogenic lines encoding each polymorphic p53 codon 72 based on induced pluripotent stem cells.
Overall, the manuscript is informative to readers, authors reported new information on global expression pattern of representative p53 target genes, polymorphic p53 under DNA damage stress, cellular signaling pathways between the polymorphic p53 codon 72, doxorubicin responsive genes, network between metabolism, p53, and DNA damage and support of their results by published studies. This study well aligns with in vivo study published by Kung et al 2017 in human p53 knock-in (Hupki) mice that suggests R72 polymorphism confers increased cell survival in response to nutrient deprivation, it holds true for the northern hemisphere where people are deprived of nutrients because of scarcity, it shows biased selection of R72 of nature. However, conclusion about differentially regulation of metabolic function by polymorphic p53 codon 72 is not new. There are several cancer research related articles published show different regulation pathways of DNA repair and apoptosis. Hu et al 2005 reported Arg/Pro polymorphism at codon 72 of the p53 gene alters the ability of the p53 protein to induce apoptosis, influences the behavior of mutant p53, decreases DNA repair capacity, and may be linked with an increased risk of lung cancer. I suggest authors should include in vivo data of Hupki mice to support their results about codon 72 polymorphism of p53 does not regulate DNA damage, growth arrest and apoptosis pathway if they are stand by their results in iPSc.
Re) According to the reviewer's good comments, we analyzed the previous data [31] additionally and reflected it in Figure 6. And the meaning of the analysis result is as follows:
In the GSE26851 data, a small number of functions were associated with core DEGs, including arachidonic acid metabolism, NF-kappa B signaling pathway, and cytokine receptor interaction pathway. Interestingly, these functions were also mentioned by the authors [31]. These results show that functional analysis using core DEGs can be useful in identifying biological differences despite differences in individual genes.
[31] Frank, A.; Leu, J. I.; Zhou, Y.; Devarajan, K.; Nedleko, T.; Klein-Szanto, A.; Hollstein, M.; Murphy, M. E., The codon 72 polymorphism of p53 regulates interaction with NFkB and transactivation of genes involved in immunity and inflammation. Mol Cell Biol 2011, 31, (6), 1201-13.
English needs to improve in the entire manuscript.
Re) We thank Editage (www.editage.co.kr) for editing and reviewing this manuscript for English language.
We appreciate your comments to improve our manuscript.
Reviewer 2 Report
In this Ms., Kim et al aimed to investigate the functional differences between the two polymorphic p53 alleles, p53-R72 and p53-P72. Using human induced pluripotent stem cells (iPSCs) derived from fibroblasts heterozygous for p53-(R72/P72) a bacterial artificial chromosome-mediated homologous recombination (BAC-mediated HR) system, the authors generated two isogenic and hemizygous fibroblast cell lines expressing either p53-R72 or p53-P72 allele. These two isogenic cell lines were then exposed to doxorubicin and processed for gene expression analysis. The authors concluded that while the two polymorphic alleles showed similar expression of genes involved in cell growth, cycle arrest and cell death/apoptosis, they exhibited different profiles for genes involved in metabolism.
General comments
The work described here has been reported previously by several groups with similar results. The only difference is the approach in generating the isogenic hemizygous cell lines for these p53 variants. There is little functional analysis in this report and the majority of the data is based on bioinformatics that is not described/explained adequately and is difficult to follow. Overall, the paper is very difficult to follow and requires significant improvements in date presentation and explanation.
Specific comments
Introduction:
- The authors should indicate the proteins encoded by the p53 target genes mentioned involved in growth arrest, apoptosis, etc.
Results:
Results are generally not well explained, and Figure legends do not adequately describe the Figures. Why does the duration of the doxorubicin treatment vary from experiment to experiment?
Figure 1:
- What are the “undifferentiated cells”? Do they mean the iPSCs? They should clarity.
- It’s confusing and not clear why the authors use *, **, #, ## symbols to indicated non-significance.
Figure 2:
- The label for the hrs of doxorubicin treatment should be in front of the numbers
- The diagrams should be explained
- The authors should clarify if the results are after 2 or 4 hours of dox treatment. Also comment on genes coordinately up or down regulated between the two isoforms and the isoforms and null phenotype
- This panel should be explained properly; what does “Functional association of temporally regulated genes from P72, R72 and p53KO cells were measured” mean? What do different colored/multi-colored circles signify? The labels are difficult to read.
Figure 3:
- The Figure needs to be explained clearly and panel by panel.
- The results from the histograms should be summarized. E.g, while both isoforms decreased metabolic pathway activity, the effect of R72 was more pronounced, etc. etc.
- explain what the correlation or lack thereof means.
Table 1:
- The authors should explain “core differential genes” and the “first neighbor analysis”
- They should indicate if there are any genes common among the 3 lines
- What does differential refer to expressed in one isotype only or expressed at different level in both isotypes.
- They should comment on the significance of genes that are expressed only in R72 or P72.
Figure 4:
- The Figure needs to be explained clearly and panel by panel.
- What is the significance of comparing the null phenotype with either isoform at 4h and the 2 isoforms at 2 and 4 hrs?
- What do different colored/multi-colored circles signify? The labels are difficult to read. Functional association between null and each isoform; what about the functional association between the 2 isoforms?
- What do the counts and colors mean? Difficult to read the labels.
- Need to explain the GO panel and the chart; difficult to read
- Each panel should be explained clearly. TP53 arrow pointing to?
Figure 5:
- The legend does not explain the Figure adequately and is hard to follow
- The heat map is missing the expression pattern for glucagon signaling pathway or the labels are not lined up, i.e., is the last row cholesterol metabolism/glucagon signaling pathway? The histogram should be explained in more detail.
Figure 6:
- The Figure needs to be explained clearly and panel by panel. The labels and writings are too small to read.
- “Pathways were classified into 6 functional categories as indicated by color below.” It’s not clear what they are referring to; there is no color below
- It’s not clear what correlations the authors refer to in the text; “However, some samples showed a positive correlation with our samples. For example, P6113 (r=0.34, p<0.001) and R5386 (r=0.36, p<0.001) were correlated with the P72 and R72 forms of our samples, respectively, as shown in the lower panel of Figure 6B.” In this statement, what are P6113 and R5386 and how would such low values (r=0.34 and r=0.36) indicate positive correlation?
- What do the ring colors represent? What is significant in this data?
- “Pathways associated with core differential genes selected from external datasets were compared between samples. Intensity in black indicates statistical significance as shown in the scale bar below.” There is no scale bar; what are they referring to?
- “In the right panel, for these pathways, the activity ratio (P72/R72) according to the P53 status was measured and compared.” The authors should describe what the results indicate and summarize the overall results presented in the Figure.
Discussion
- It’s not clear what that the authors are trying to say in the following statements:
“By using neighboring genes directly interacting with an interesting gene, we applied a novel analytic strategy to investigate signaling pathways that could be differentially regulated by P72 or R72. The neighboring gene analytic approach precisely means to include adjacent genes that interact directly with the gene of interest in an interaction network and analyze their expression patterns. The analytic approach measures the expression levels of genes that are functionally associated with P72 or R72 to isolate core differential genes. By comparing P72 and R72, it was identified that the expression of metabolic-related gene groups among functional categories was significantly different.”
“Moreover, P72 and R72 were markedly reduced by R72 in 305 pathways belonging to the functional category of metabolism, especially in lipid-related 306 metabolism.”
- In discussing and comparing Kung et al results with the results presented here the authors stated, “The reason we could not confirm the difference in the expression of the two genes in the P72 and R72 cell lines can be seen as a result of the difference in species in the studies.” This logic is unlikely since Kung et al. used a humanized mouse model and assessed the functional differences in homozygous P72 and R72 in detail.
- In comparing their current results with previous studies, the authors stated, “Paradoxically, R72, which is associated with metabolic disease, had a significantly lowered effect on the expression of metabolic disease-related genes under genotoxic stress. This suggests that the pathway shift by p53 protein, especially R72, between metabolic signaling pathway and DNA damage pathway under genotoxic stress, can have a greater effect.” It’s not clear what the authors mean by these statements specially the last sentence.
Author Response
Response to Reviewer 2 Comments
In this Ms., Kim et al aimed to investigate the functional differences between the two polymorphic p53 alleles, p53-R72 and p53-P72. Using human induced pluripotent stem cells (iPSCs) derived from fibroblasts heterozygous for p53-(R72/P72) a bacterial artificial chromosome-mediated homologous recombination (BAC-mediated HR) system, the authors generated two isogenic and hemizygous fibroblast cell lines expressing either p53-R72 or p53-P72 allele. These two isogenic cell lines were then exposed to doxorubicin and processed for gene expression analysis. The authors concluded that while the two polymorphic alleles showed similar expression of genes involved in cell growth, cycle arrest and cell death/apoptosis, they exhibited different profiles for genes involved in metabolism.
General comments
The work described here has been reported previously by several groups with similar results. The only difference is the approach in generating the isogenic hemizygous cell lines for these p53 variants. There is little functional analysis in this report and the majority of the data is based on bioinformatics that is not described/explained adequately and is difficult to follow. Overall, the paper is very difficult to follow and requires significant improvements in date presentation and explanation.
Re) Sorry for the inconvenience. According to reviewer’s opinions, we intensively revised the manuscript like below.
Specific comments
Introduction:
- The authors should indicate the proteins encoded by the p53 target genes mentioned involved in growth arrest, apoptosis, etc.
Re) We represented the proteins encoded by the p53 target genes in the introduction section.
Results:
- Results are generally not well explained, and Figure legends do not adequately describe the Figures. Why does the duration of the doxorubicin treatment vary from experiment to experiment?
Re) To identify the expression of p53 target genes, we set the duration of the doxorubicin treatment from 2 to 6 hours Within this time range, the difference in p53-dependent gene expression pattern by drugs can be confirmed. And in iPSCs, doxorubicin-induced cell death appeared 6 hours later. Therefore, we determined the apoptosis assay condition to be 6 hours after drug treatment. And, to confirm the difference in growth arrest caused by polymorphic p53 codon 72, we measured the difference in the degree of EdU incorporation during cell division, that is, DNA synthesis. In the case of endodermal lineage, the doubling time occurs 12 hours later. Thus, after treating the drug for 12 hours, the difference in growth arrest was confirmed by culturing for an additional 16 hours in EdU-added medium. All experiments were performed based on previously reported conditions (Oncogene 38, 1597-1610 (2019)).
Figure 1:
- What are the “undifferentiated cells”? Do they mean the iPSCs? They should clarity.
Re) Undifferentiated cells are represented as iPSCs for clarity in the result section and Figure 1 legend.
- It’s confusing and not clear why the authors use *, **, #, ## symbols to indicated non-significance.
Re) We removed the symbols that could be confusing in the Figure 1 or legend.
Figure 2:
- The label for the hrs of doxorubicin treatment should be in front of the numbers.
Re) According to reviewer’s comment, we labeled it.
- The diagrams should be explained.
- The authors should clarify if the results are after 2 or 4 hours of dox treatment. Also comment on genes coordinately up or down regulated between the two isoforms and the isoforms and null phenotype.
Re) The above two comments have been explained below:
“As shown in Figure 2C, unlike p53 null cells, a large number of genes (154 temporally up-regulated genes and 9 temporally down-regulated genes) in P72 and R72 cells responded in common at 4 h after doxorubicin treatment.”.
- This panel should be explained properly; what does “Functional association of temporally regulated genes from P72, R72 and p53KO cells were measured” mean? What do different colored/multi-colored circles signify? The labels are difficult to read.
Re) According to reviewer’s comments, we explained it like below, and represented the meaning of the colors in the Figure 2D legend.
“We then measured biological functions associated with genes temporally regulated by doxorubicin. As shown in Figure 2D, genes in p53 null cells were associated with nucleosome assembly, whereas genes in P72 or R72 cells were associated with cell cycle arrest, drug response, and DNA damage response. This result is consistent with the fact that there were many common genes between the genes isolated from the two cells (Figure 2B) and proves the role of doxorubicin as a DNA damaging agent.”.
Figure 3:
- The Figure needs to be explained clearly and panel by panel.
Re) Sorry for inconvenience. according to reviewer’s comments, the Figures are explained clearly and panel by panel.
- The results from the histograms should be summarized. E.g, while both isoforms decreased metabolic pathway activity, the effect of R72 was more pronounced, etc. etc.
Re) The results about histograms were described like below:
“The overall pattern of pathway activity showed a close correlation (r=0.77, p<0.001) between P72 and R72 cells, but a low correlation between P72 (or R72) and p53 null cells (Figure 3B). Categorical classification of pathways showed that in P72 or R72 cells pathways included in the metabolic and genetic information processing categories were downregulated, whereas pathways in the signaling category were upregulated by doxorubicin. For example, the activities of the citrate cycle and apoptosis pathway were significantly downregulated and upregulated, respectively, in both the P72 and R72 cells when compared with p53 null cells.”.
- explain what the correlation or lack thereof means.
Re) We explained the meaning of the correlation like below:
Although the overall pathway activity distribution was similar between P72 and R72 cells, more detailed pathway category analysis confirmed that some specific functions were different between P72 and R72 cells. In particular, it was determined that the activity of pathways involved in metabolic function was lower in R72 (Figure 3C).
Table 1:
- The authors should explain “core differential genes” and the “first neighbor analysis”.
Re) We explained it like below:
“We applied a novel analytical approach that used interaction networks of genes to iso-late genes differentially regulated between two experimental conditions. Specifically, a module consisting of the first neighbor interacting with each gene of interest was isolated from the integrative interaction network. Then, the expression levels of the genes included in the module were used to determine the expressional differences, expressional deviations, correlation coefficients between the two experiments, and the statis-tical significance was determined by comparing it with the results from the randomly selected modules from two experiments. Finally, modules showing empirical p<0.01 were isolated, and the core genes of each module were called core differentially expressed genes (core DEGs).”.
- They should indicate if there are any genes common among the 3 lines.
Re) We listed them in Table 1.
- What does differential refer to expressed in one isotype only or expressed at different level in both isotypes.
Re) Sorry for inconvenience. We clearly represented it in Table 1, marking p53 null-P72, p53 null-R72, or P72-R72.
- They should comment on the significance of genes that are expressed only in R72 or P72.
Re) We commented it like below.
“As shown in left panel of Figure 4A, 325 and 320 core DEGs were identified between p53 null cells and P72 or R72 cells at 4 h after doxorubicin treatment, respectively. The high number of common core DEGs (144 genes) among them indicates similarities between P72 and R72 cells compared to p53 null cells. On the other hands, 129 and 339 core DEGs were isolated between the P72 and R72 cells at 2 h and 4 h after doxorubicin treatment, respectively (right panel of Figure 4A). The increase in core DEGs (from 129 to 339 genes) according to doxorubicin treatment time and a relatively small number of common core DEGs (24 genes) among them showed that the difference between P72 and R72 increased with time after doxorubicin treatment. The selected list of core DEGs is shown in Table 1.”.
Figure 4:
- The Figure needs to be explained clearly and panel by panel.
Re) Sorry for inconvenience. according to reviewer’s comments, the Figures are explained clearly and panel by panel.
- What is the significance of comparing the null phenotype with either isoform at 4h and the 2 isoforms at 2 and 4 hrs?
Re) As we have already responded to the comment above, the reason we compared the left panel or right panel in Figure 4A is as follows:
“As shown in left panel of Figure 4A, 325 and 320 core DEGs were identified be-tween p53 null cells and P72 or R72 cells at 4 h after doxorubicin treatment, respectively. The high number of common core DEGs (144 genes) among them indicates similarities between P72 and R72 cells compared to p53 null cells. On the other hands, 129 and 339 core DEGs were isolated between the P72 and R72 cells at 2 h and 4 h after doxorubicin treatment, respectively (right panel of Figure 4A). The increase in core DEGs (from 129 to 339 genes) according to doxorubicin treatment time and a relatively small number of common core DEGs (24 genes) among them showed that the difference between P72 and R72 increased with time after doxorubicin treatment.”.
- What do different colored/multi-colored circles signify? The labels are difficult to read. Functional association between null and each isoform; what about the functional association between the 2 isoforms?
Re 1) We described the meaning of different colored/multi-colored circles like below:
“Association of core DEGs between P72 (or R72) and p53 null cells with GO-term was measured using ClueGO program to make a GO-term network. Each node represents a GO term, and the node size is proportional to the number of genes associated with the GO-term. Closely related GO-terms are marked with the same color. Representative GO-terms are highlighted.”.
Re 2) Functional association between null and each isoform, or between two isoforms has been described as follows:
“Measurement of functional associations for these core DEGs between p53 null cells and P72 or R72 cells at 4 h after doxorubicin treatment showed that p53-dependent functions such as cell cycle, apoptosis, and DNA damage response were significantly enriched by doxorubicin treatment (Figure 4B). These results are consistent with the generally known function of p53 on doxorubicin, thus demonstrating that the net-work-based gene selection method used in this study could be very useful for measuring biological changes. In contrast, core DEGs between p72 and R72 cells, which represent genes differentially expressed between P72 and R72 cells in response to doxorubicin (4 h after treatment) were associated with metabolism, particularly lipid-related metabolism (False discover rate (FDR) < 0.01) in pathway enrichment analysis as shown in upper panel of Figure 4C.”.
- What do the counts and colors mean? Difficult to read the labels.
Re) As can seen from the Figure 4 C legend, it indicated as follows:
In the pathway analysis, the distribution of false discovery rate (FDR) values of each pathway and the number of genes included in the pathway were shown according to the ratio of genes included in the pathway.
- Need to explain the GO panel and the chart; difficult to read
Re) It has been explained as follows:
In the GO-term analysis (lower panel), ClueGO program was used to construct an enriched GO-term network.
- Each panel should be explained clearly. TP53 arrow pointing to?
Re) Each panel has been explained in the Figure 4D legend as follows:
“The location of each core DEGs between P72 and R72 cells was marked in the interaction network. Directly connected core DEGs were indicated in the right panel, and the color of the nodes indicates the number of nodes connected to each node as shown in scale bar. The positon of p53 genes was indicated.”.
Figure 5:
- The legend does not explain the Figure adequately and is hard to follow.
Re) Sorry for inconvenience. We clearly described it for Figure 5 legend.
- The heat map is missing the expression pattern for glucagon signaling pathway or the labels are not lined up, i.e., is the last row cholesterol metabolism/glucagon signaling pathway? The histogram should be explained in more detail.
Re) Thanks for your comment. The mismatch between line and label was remarked in the Figure 5C, explaining it in the Figure 5C legend.
Figure 6:
- The Figure needs to be explained clearly and panel by panel. The labels and writings are too small to read.
Re) Sorry for inconvenience. We described it in detail.
- “Pathways were classified into 6 functional categories as indicated by color below.” It’s not clear what they are referring to; there is no color below.
Re) It has been represented in the Figure 6A legend as follows:
“Six functional categories including each pathway are indicated by color next to the figure. In GSE61124, P72 or R72 was present for each of the two cell types, respectively.”.
- It’s not clear what correlations the authors refer to in the text; “However, some samples showed a positive correlation with our samples. For example, P6113 (r=0.34, p<0.001) and R5386 (r=0.36, p<0.001) were correlated with the P72 and R72 forms of our samples, respectively, as shown in the lower panel of Figure 6B.” In this statement, what are P6113 and R5386 and how would such low values (r=0.34 and r=0.36) indicate positive correlation?
Re) Sorry for inconvenience. External public datasets are similar to our data in terms of overall metabolism, but external public datasets for 6 pathways, including steroid biosynthesis, terpenoid backbone biosynthesis, ferroptosis, porphyrin and chlorophyll metabolism, cholesterol metabolism, and glucagon signaling shown in the Figure 5C, are not correlation with our data. Thus, the lower panel of Figure 6B and the text describing it have been deleted.
- What do the ring colors represent? What is significant in this data?
Re 1) The meaning of ring color has been represented in the Figure 6C legend as follows:
“The distribution of core DEGs selected from the external data sets was compared with those selected from our P72 and R72 cells. In GSE61124, 4 types of core DGEs were isolated from P72 and R72 at 4 hours after stimulation for each cell type, and in GSE26851, 2 types of core DGEs were isolated for each cell type. The core DEGs in our experiments are highlighted in color.”.
Re 2) We conferred the meaning of them in the result section as follows:
“Figure 6C shows the distribution of these core DEGs between the external samples and our sample. There were some core DEGs in common between our experimental sample and external data, but not many. Therefore, this time, we measured whether the functions associated with core DEGs are similar between samples.”.
- “Pathways associated with core differential genes selected from external datasets were compared between samples. Intensity in black indicates statistical significance as shown in the scale bar below.” There is no scale bar; what are they referring to?
Re) We added it in the Figure 6D. The cut-off of 1E-3 means a statistical value with significance. That is, Intensity in black indicates statistical significance.
- “In the right panel, for these pathways, the activity ratio (P72/R72) according to the P53 status was measured and compared.” The authors should describe what the results indicate and summarize the overall results presented in the Figure.
Re) Sorry for inconvenience. As we have already described the reason the low panel of Figure 6B and related contents were deleted, for the same reason, we deleted Figure 6E and related contents. When analyzed based on core DEGs of the Figure 6C, our six pathways, including steroid biosynthesis, terpenoid backbone biosynthesis, ferroptosis, porphyrin and chlorophyll metabolism, cholesterol metabolism, and glucagon signaling, were not correlation with external public datasets. These results may be reflected in differences between samples.
Discussion
- It’s not clear what that the authors are trying to say in the following statements:
“By using neighboring genes directly interacting with an interesting gene, we applied a novel analytic strategy to investigate signaling pathways that could be differentially regulated by P72 or R72. The neighboring gene analytic approach precisely means to include adjacent genes that interact directly with the gene of interest in an interaction network and analyze their expression patterns. The analytic approach measures the expression levels of genes that are functionally associated with P72 or R72 to isolate core differential genes. By comparing P72 and R72, it was identified that the expression of metabolic-related gene groups among functional categories was significantly different.”
- “Moreover, P72 and R72 were markedly reduced by R72 in 305 pathways belonging to the functional category of metabolism, especially in lipid-related 306 metabolism.”
Re) For clarity, one paragraph in the discussion section has included the contents of two sentences commented by the reviewer and has been updated as follows:
“The polymorphic p53 codon 72 has a characteristic as a non-conservative single nucleotide change, but the functional similarity between P72 and R72, such as cell cycle arrest and apoptosis, is so close that it was somewhat difficult to clearly identify the functional difference according to the polymorphism. Therefore, we applied the gene interaction network, the so-called network-based gene selection method, to isolate genes differentially regulated by the polymorphic p53 codon 72. The useful value of this analytical method was confirmed by the apparent functional difference between p53 null and polymorphic p53 codon 72, such as cell cycle arrest, apoptosis, or DNA damage response. Through network-based gene selection method, we could identify that core DEGs representing genes differentially expressed between P72 and R72 cells in response to doxorubicin were associated with metabolism, particularly lipid-related metabolism.”.
- In discussing and comparing Kung et al results with the results presented here the authors stated, “The reason we could not confirm the difference in the expression of the two genes in the P72 and R72 cell lines can be seen as a result of the difference in species in the studies.” This logic is unlikely since Kung et al. used a humanized mouse model and assessed the functional differences in homozygous P72 and R72 in detail.
Re) We modified the sentence like below, then described it in the discussion section.
“The reason we could not confirm the difference in the expression of the two genes in the P72 and R72 cell lines can be seen as a result of the difference in experimental conditions. That is, unlike our genotoxic stress, the previous study produced data from HFD-fed mouse.”
- In comparing their current results with previous studies, the authors stated, “Paradoxically, R72, which is associated with metabolic disease, had a significantly lowered effect on the expression of metabolic disease-related genes under genotoxic stress. This suggests that the pathway shift by p53 protein, especially R72, between metabolic signaling pathway and DNA damage pathway under genotoxic stress, can have a greater effect.” It’s not clear what the authors mean by these statements specially the last sentence.
Re) For clarity, the last sentence has been replaced with the sentence below, then described it in the discussion section.
“As shown in Figure 2B, the expression of metabolism-related genes was significantly reduced in both P72 and R72 cells under DNA damage conditions, and in particular, the expression of these genes showed a tendency to further decrease in R72 cells. Although further experimental confirmation is required, this suggests that metabolic dysfunction may be relatively vulnerable in R72 cells through the reduction of metabolism-related genes, which is more marked in R72 than in P72.”
We appreciate your comments to improve our manuscript.
Reviewer 3 Report
The TP53 gene was one of the earliest genes, whose mutations were associated with cancer. Different studies on mutant TP53 have revealed diverse pro-tumorigenic mechanisms that occur in response to the mutations. At codon 72 in the TP53 gene, a polymorphism is reported which has been associated with various cancers. The polymorphism results in the codon 72 to encode either for Arg or for Pro. This causes functional impact on p53 itself.
In this study, the authors have generated isogenic lines from human induced Pluripotent stem cells (iPSCs) that encode either Arg or Pro at TP53 gene codon 72. The isogenic lines express the corresponding amino acid containing protein from one allele of the genome. The other allele is silenced by incorporating a BAC construct containing a premature STOP codon.
Comment 1:
The novelty/originality of this study has been the generation of the isogenic cell-line from human iPSCs. In humans the codon 72 polymorphism results in Pro/Pro, Pro/Arg or Arg/Arg expression from codon 72. However, the isogenic line developed in this study consists of Pro/-, Arg/- or -/- configuration at codon 72. How closely does this reflect the patient setting is not explained or shown.
Comment 2:
The authors use one method for causing DNA damage – Doxorubicin. But the confirmation of the DNA damage is not shown by using specific staining for DNA damage markers such as ϒ-H2AX is not shown. That raises the question as to whether the experimental setup accurately represents the intended conditions for subsequent experiments.
Comment 3:
The authors acknowledge that the gene expression data in this study, at some occasions, contradicts other similar studies where gene expression was analyzed.
Overall, this study makes interesting observations using gene expression analysis and the comparison thereof to other existing expression datsasets.
Author Response
Response to Reviewer 3 Comments
The TP53 gene was one of the earliest genes, whose mutations were associated with cancer. Different studies on mutant TP53 have revealed diverse pro-tumorigenic mechanisms that occur in response to the mutations. At codon 72 in the TP53 gene, a polymorphism is reported which has been associated with various cancers. The polymorphism results in the codon 72 to encode either for Arg or for Pro. This causes functional impact on p53 itself.
In this study, the authors have generated isogenic lines from human induced Pluripotent stem cells (iPSCs) that encode either Arg or Pro at TP53 gene codon 72. The isogenic lines express the corresponding amino acid containing protein from one allele of the genome. The other allele is silenced by incorporating a BAC construct containing a premature STOP codon.
Comment 1:
The novelty/originality of this study has been the generation of the isogenic cell-line from human iPSCs. In humans the codon 72 polymorphism results in Pro/Pro, Pro/Arg or Arg/Arg expression from codon 72. However, the isogenic line developed in this study consists of Pro/-, Arg/- or -/- configuration at codon 72. How closely does this reflect the patient setting is not explained or shown.
Re) According to the reviewer's good comment, we described the following at the bottom of the first paragraph of the discussion section.
“Although only another functional copy is retained by removing one copy of the polymorphic p53 codon 72, respectively, the retention of another copy can apparently reflect on the functional differences between isogenic lines. Also, our isogenic line retains heterozygosity of p53, thus It may provide valuable research for Li-Fraumeni syndrome, including loss of heterozygosity at the p53 gene locus, as an inherited p53 mutation.”
Comment 2:
The authors use one method for causing DNA damage – Doxorubicin. But the confirmation of the DNA damage is not shown by using specific staining for DNA damage markers such as ϒ-H2AX is not shown. That raises the question as to whether the experimental setup accurately represents the intended conditions for subsequent experiments.
Re) Thanks for pointing out what we missed. We added immno-blotting data for DNA damage marker, including phosphor-γH2AX (ser139), to figure 1B.
Comment 3:
The authors acknowledge that the gene expression data in this study, at some occasions, contradicts other similar studies where gene expression was analyzed.
Overall, this study makes interesting observations using gene expression analysis and the comparison thereof to other existing expression datsasets.
Re) We appreciated your good suggestions and comments.
Round 2
Reviewer 1 Report
Dear Authors,
I reviewed your improved manuscript, I agree Frank et al (2011) mentioned about it. However, I would like to see to include more published research articles in discussion since there are several reports by many groups. DEGs in p72 or r72 eventually make people vulnerable to cancers (Shen et al 2003). Authors should discuss DEGs functions in long time. I would be willing to see this manuscript once again once the basic aspects for publication have been established as mentioned above.
Author Response
I reviewed your improved manuscript, I agree Frank et al (2011) mentioned about it. However, I would like to see to include more published research articles in discussion since there are several reports by many groups. DEGs in p72 or r72 eventually make people vulnerable to cancers (Shen et al 2003). Authors should discuss DEGs functions in long time. I would be willing to see this manuscript once again once the basic aspects for publication have been established as mentioned above.
Re) According to the reviewer's good comments, we described it in the discussion section as follows:
“Some studies have observed no association between the polymorphic p53 codon 72 and cancer prevention, which support this assumption [23-29]. Conclusively, each polymorphic p53 codon 72 mice retaining P72 or R72, respectively, were not able to determine which type was more prone to cancer, but also showed no difference in survival rate [33]. Exploring the correlation with the expression of p53 target genes depending on polymorphic types in the mouse-derived embryonic fibroblast, the expression of Cdkn1a, a gene related to growth arrest, was relatively increased by P72 [33]. This result suggests cell cycle arrest that is more strongly regulated by P72, consistent with previous results [13, 14]. However, incompatible with previous reports [11-13], there was no significant difference in the expression of proapoptotic genes including Puma, Noxa, and Bax by R72, comparing with P72 [33].
On the other hand, some studies have reported that one of the polymorphic p53 codon 72 has been predisposed to tumor formation [18-22]. R72 was enriched in human papilloma virus-positive cervical cancers [18-20]. It was also significantly associated with an increased risk of developing melanoma [21]. In addition to R72, P72 is known to induce tumor promotion by increasing cell survival and epithelial-mesenchymal transition, and consequently contribute to the aggressiveness of colorectal cancer, inducing activation of p38 and RAF/MEK/ERK pathway [22].”.
We appreciate your comment to improve our manuscript.
